# *SERPINE1* DNA Methylation Levels Quantified in Blood Cells at Five Years of Age Are Associated with Adiposity and Plasma PAI-1 Levels at Five Years of Age

**DOI:** 10.3390/ijms231911833

**Published:** 2022-10-05

**Authors:** Amelie Taschereau, Véronique Desgagné, Sabrina Faleschini, Renée Guérin, Catherine Allard, Patrice Perron, Marie-France Hivert, Luigi Bouchard

**Affiliations:** 1Department of Biochemistry and Functional Genomics, Faculty of Medicine and Health Sciences (FMHS), Université de Sherbrooke, Sherbrooke, QC J1K 2R1, Canada; 2Clinical Department of Laboratory Medicine, Centre Intégré Universitaire de Santé et de Services Sociaux du Saguenay–Lac-St-Jean, Saguenay, QC G7H 7K9, Canada; 3Département D’études sur L’enseignement et L’apprentissage, Faculté des Sciences de L’éducation, Université Laval, Québec, QC G1V 0A6, Canada; 4Centre de Recherche du Centre Hospitalier Universitaire de Sherbrooke, Universitaire de Sherbrooke, Sherbrooke, QC J1H 5N4, Canada; 5Département de Médecine, Faculty of Medicine and Health Sciences (FMHS), Université de Sherbrooke, Sherbrooke, QC J1K 2R1, Canada; 6Department of Population Medicine, Harvard Pilgrim Health Care Institute, Harvard Medical School, Boston, MA 02115, USA; 7Diabetes Unit, Massachusetts General Hospital, Boston, MA 02114, USA

**Keywords:** obesity, PAI-1, epigenetics, childhood

## Abstract

Plasminogen activator inhibitor (PAI-1) expression has been associated with a higher risk of development of obesity. DNA methylation (DNAm) is an epigenetic mechanism regulating gene transcription and likely involved in the fetal programming of childhood obesity. Our study aimed to assess the associations between PAI-1 gene (*SERPINE1*) DNAm, plasma PAI-1 levels, and adiposity at five years of age. We analyzed DNAm and anthropometric data from 146 girls and 177 boys from the Gen3G prospective birth cohort. We assessed adiposity using BMI z-scores, waist circumference, total skinfolds, and percentages of total, android, and trunk fat measured by dual-energy radiography (DXA). We estimated blood cell DNAm levels at 15 CpG sites within *SERPINE1* using the methylationEPIC array. After correction for multiple testing, we found that lower DNAm in *SERPINE1* intron 3 (cg11353706) was associated with greater adiposity levels in girls (waist circumference: r = −0.258, *p* = 0.002; skinfolds: r = −0.212, *p* = 0. 013; android fat: r = −0.215, *p* = 0.015; BMI z-score: r = −0.278, *p* < 0.001) and that lower DNAm in the *SERPINE1* promoter (cg19722814) was associated with higher plasma PAI-1 levels in boys (r = −0.178, *p* = 0.021). Our study suggests that DNAm levels at the *SERPINE1* gene locus are negatively correlated with adiposity, but not with plasma PAI-1 levels, in young girls only.

## 1. Introduction

The prevalence of childhood obesity has increased dramatically over the past few decades, now affecting 39 million children aged five and below worldwide (2020) [1]. Obesity is defined as an excessive accumulation of fat mass that can affect health [1]. To classify adiposity levels, the World Health Organization (WHO) recommends the use of the body mass index (BMI), which consists of the measurement of weight in kilograms (kg) divided by the square of the height in meters (m). In children aged 5 to 19 years, the WHO recommends the use of BMI z-scores, which are BMI measures adjusted for age and sex [1]. The threshold for childhood obesity consists of the 97th percentile BMI by age group and sex, which is equivalent to a BMI z-score of 2 [2]. Development of obesity at an early age is associated with an increased risk of metabolic, cardiovascular, and liver diseases earlier in life [3], and an increased risk of obesity in adulthood [4]. A better understanding of the mechanisms underlying its development is therefore urgently needed to develop better and updated preventive approaches. 

Plasminogen activator inhibitor (PAI-1) is a glycoprotein primarily known to regulate fibrinolysis by inhibiting both tissue-type and urokinase-type plasminogen activators (t-Pa and u-Pa) [5]. PAI-1 is secreted by adipocytes, among other cell types [6], and its expression is upregulated in the contexts of obesity [7] and insulin resistance [8,9]. High plasma PAI-1 levels are associated with an increased cardiovascular risk in obese individuals due to the glycoprotein’s pro-thrombotic effect and the risk factors it represents [10]. However, rather than just being a marker of obesity and insulin resistance states, PAI-1 could also be implicated in pathophysiological mechanisms underlying their development.

Indeed, PAI-1 has been associated with insulin signaling impairments by competing with αvβ3 integrin for vitronectin binding, which is required for complete activation of insulin receptor substrate (IRS) [11]. In addition, due to its inhibitory effect on t-Pa and u-Pa, which are implicated in extracellular matrix degradation, PAI-1 has been associated with decreased angiogenesis in adipose tissue [12], which could promote development of this tissue by inhibiting the release of fatty acids [8]. Moreover, studies conducted with diet-induced obesity mice models showed a protective effect of both pharmacological inhibition, using an inhibitor of PAI-1 (PAI-039), and genetic deletion of PAI-1 against the development of both obesity and insulin resistance [13,14]. A decrease in the infiltration of pro-inflammatory macrophages into adipose tissue has also been reported [15,16]. Despite conflicting results in the literature [17,18,19], these data suggest a possible causal role for PAI-1 in the development of obesity.

Genetic, epigenetic, and transcriptomic studies also support the role of PAI-1 in the development of obesity and insulin resistance. For instance, the 4G allele of the 4G/5G polymorphism of the *SERPINE1* gene, which is associated with increased gene transcription of the glycoprotein, was overrepresented in obese adults as compared to the 5G allele that is not associated with elevated transcription [20]. Lopez-Legarrea et al. also reported associations between DNA methylation (DNAm) levels in venous blood samples at the promoter region of the *SERPINE1* gene and anthropometric and metabolic changes in obese adults undergoing a calorie-restricted diet [21]. DNAm consists of the addition of a methyl group to the fifth carbon of a cytosine which is followed by guanine [22], which is well known to regulate gene transcription. DNAm is an epigenetic modification that is also central to the concept of programming chronic diseases such as obesity [23]. However, no study has investigated the association between DNAm levels at the *SERPINE1* locus and adiposity in childhood, a critical period for metabolic health programming.

We thus hypothesized that blood cell DNAm variations at the *SERPINE1* gene locus contribute to the regulation of the expression of PAI-1 and the development of obesity in childhood. To test our hypothesis, we first assessed the association between plasma PAI-1 levels and different markers of adiposity at 5 years of age. We then investigated the association between *SERPINE1* DNAm levels in blood cells and adiposity markers and plasma PAI-1 levels at 5 years of age. 

## 2. Results

### 2.1. Participant Characteristics

The characteristics of the 341 children selected for this study are shown in Table 1. Briefly, girls and boys were ≈64.2 months old on average. Mean (±standard deviation (SD)) PAI-1 plasma levels were higher in girls compared to boys (9.99 ± 12.29 pg/mL vs. 7.84 ± 6.39 pg/mL; *p* = 0.049). Girls had a mean BMI z-score of 0.279 ± 0.971, skinfold thickness of 35.47 ± 11.99 mm, and waist circumference of 54.61 ± 4.79 cm. In girls, the mean percent trunk, android, and total fat measured by DXA was 28.81 ± 4.60%, 29.03 ± 5.36%, and 33.06 ± 4.33%, respectively. Boys had a similar mean BMI z-score (0.188 ± 0.932; *p* = 0.382) as compared to girls. However, they had lower mean skinfold thickness (28.01 ± 7.77 mm; *p* < 0.001), waist circumference (53.65 ± 3.44 cm; *p* = 0.038), and percentages of trunk, android, and total fat measured by DXA (23.89 ± 3.03%, 24.11 ± 3.26%, and 28.46 ± 3.32%, respectively; *p* < 0.001).

### 2.2. Genomic Context of the SERPINE1 Gene 

The human *SERPINE1* gene is localized on chromosome 7q22.1. The 15 CpG sites analyzed are shown in their genomic context, including the indexed transcription factor binding sites (TFBS), in Figure 1. Their genomic locations are reported in Appendix A. The co-correlations between DNAm levels at each of the CpG sites analyzed were also tested. Overall, only weak correlations have been observed (Appendix A).

The figure shows the *SERPINE1* gene locus, its transcript, and the alternative BX649164 transcript (chr7: 100,774,814 [24]). The 15 CpG sites analyzed, and their “CG” identification number are also represented. Exons are shown as black boxes. Transcription factor binding sites are also shown: three hypoxia-responsive elements (HRE; −452, −195, and −161 from the TSS), an estrogen response element (ERE at −427 to −407), one SMAD protein-binding site mediating TGF-β responsiveness (Smad at −280) and a p53 responsiveness element (P53 at −159 to −134) in the same region of a binding site for a PAI-1 negative regulator (E2F) [25]. 

### 2.3. Associations between Plasma PAI-1 Levels and Adiposity Levels at 5 Years of Age

We first tested the cross-sectional associations between plasma PAI-1 levels and adiposity markers in all children together. Only trends toward associations were obtained with waist circumference and percentages of trunk and total fat (r = 0.095, *p* = 0.081; r = 0.108, *p* = 0.059; r = 0.104, *p* = 0.070, respectively) (Appendix A). We then tested for an interaction term between plasma PAI-1 levels and sex, which was assessed for each of the adiposity marker dependent variables (Appendix A). Because no interaction term reached the significance threshold (*p* < 0.05), the analyses were not repeated in boys and girls separately.

### 2.4. Associations between Blood Cell DNAm Levels at the SERPINE1 Gene Locus and Plasma PAI-1 Levels at 5 Years of Age

We then tested the cross-sectional associations between blood cell DNAm levels and plasma PAI-1 levels in all children together but none of the associations reached the threshold for significance (Appendix A). However, we also tested for an interaction term between DNAm and sex at each of the annotated CpG sites for the dependent variable of plasma PAI-1 levels (Appendix A). A significant interaction with sex was observed for cg19722814 and the analysis was repeated in both sexes separately for this CpG site. We obtained a negative correlation between DNAm levels at cg19722814 and plasma PAI-1 levels in boys only (r = −0.178, *p* = 0.021) (Figure 2, Appendix A). Interestingly, this CpG site is downstream of a hypoxia-responsive element (HRE) and upstream of an estrogen-responsive element (ERE) [25] (Figure 1).

### 2.5. Associations between Blood Cell DNAm Levels at the SERPINE1 Gene Locus and Adiposity Levels at 5 Years of Age

We then tested the cross-sectional associations between blood cell DNAm levels and adiposity markers in all children but none of them reached the threshold for significance when boys and girls were considered together in the analyses (Appendix A). However, we observed significant interactions between sex and four CpG sites (cg25826546, cg02273392, cg11353706, and cg01975495; Appendix A) for these models. These associations were therefore retested in girls and boys separately. After correction for age, batch effect, and cell-type heterogeneity, three CpG sites (cg25826546, cg11353706, cg01975495) correlated significantly with adiposity markers in girls, as compared to only one (cg02273392) in boys (Appendix A). After correction for multiple testing (*p* < 0.017), only cg11353706 remained significantly associated with adiposity markers in girls, including waist circumference (r = −0.258, *p* = 0.002), percent android fat mass (r = −0.215, *p* = 0.015), skinfold thickness (r = −0.212, *p* = 0.013) and BMI z-score (r = −0.278, *p* < 0.001), (Figure 3, Appendix A).

Interestingly, cg11353706 is located within the third intron of the *SERPINE1* gene and is only a few nucleotides upstream of the transcription start site of an mRNA known as BX649164 [24] (Figure 1).

Considering specifically the CpG sites associated with adiposity, before correction for multiple testing, DNAm levels at cg25826546 were partially correlated with those at cg11353706 (r = 0.267, *p* = 0.001) and cg01975495 (r = 0.221, *p* = 0.007) in girls only, whereas DNAm levels at cg11353706 and cg01975495 were correlated with each other in girls (r = 0.221, *p* = 0.007) and boys (r = 0.209, *p* = 0.005). Interestingly, DNAm levels at cg02273392 did not correlate with any of the DNAm levels measured at cg25826546, cg11353706, or cg01975495 in either girls or boys (Supplementary Appendix A).

## 3. Discussion

Growing evidence suggests that PAI-1 may have a causal role in adipogenesis. For instance, regulatory mechanisms involved in the differentiation of PAI-1 expression are associated with obesity in humans [19,20]. On the other hand, it is well established that altered expression of genes through epigenetic modification can lead to adiposity dysfunction and the development of obesity [26]. Our study reports correlations between blood cell DNAm levels at the *SERPINE1* locus and adiposity markers as well as plasma PAI-1 levels in children aged approximately five years. 

Firstly, we observed a negative correlation between blood cell DNAm levels at the cg19722814 locus, located in the promoter region of the *SERPINE1* gene, and plasma PAI-1 levels in boys only. Although this association is modest, this result is consistent with those obtained by Gao et al., who reported a negative correlation between the degree of methylation of the *SERPINE1* 5’-flanking region and PAI-1 mRNA levels in human cell lines [25]. One mechanism by which methylation in the promoter region can decrease gene transcription is by preventing the binding of transcription factors to the DNA sequence [27]. Interestingly, cg19722814 is located (Genome assembly: GRCh37 (CGA:100769933)) [24] only a few nucleotides downstream of the beginning of a transcription factor binding site for estrogen (estrogen response element or ERE) which promotes *SERPINE1* transcriptional activity [25]. Estrogen is detectable in blood circulation in prepubertal boys, although its concentration is about eight times lower than that in girls, with far fewer interindividual differences [28]. Accordingly, the potential role of this ERE in boys remains unclear but might contribute to explaining the increased PAI-1 levels in girls even when cg19722814 is methylated. 

Secondly, our study reports negative correlations between blood cell DNAm and adiposity only in girls. Despite the fact that they did not consider sex in their analysis, our results are consistent with those obtained by Lopez-Legarrea et al. [21]. However, although we report associations in the same direction (i.e., negatively correlated), the CpG sites identified are in different genomic contexts: gene body (cg11353706) vs. promoter (cg20583316). Although DNAm in the promoter region very often results in transcriptional repression, its impact when it occurs in the gene body is more complex and will depend on the genomic context that hosts the modification [29]. Among others, the literature currently demonstrates an implication of intronic DNAm in the regulation of tissue-specific alternative transcript expression [30,31]. Interestingly, cg11353706 is within the third intron of *SERPINE1*, only three nucleotides upstream of the transcriptional start site of an mRNA (BX649164) that is encoded in the middle of *SERPINE1* and in a DNase hypersensitive site (Genome assembly: GRCh37 (CGA100,774,811)) [24]. The associations we observed between methylation levels at the cg11353706 locus and adiposity in girls could be explained by changes in BX649164 expression in girls. Unfortunately, the function of BX649164 remains unknown to date. Fine mapping DNAm and transcriptomic analyses of this *SERPINE1*-specific locus are thus needed to identify the CpG site(s) that is most likely responsible for the association with body fat in girls at 5 years of age, and its impact on *SERPINE1*/BX649164 transcriptional activities. 

Although we did find correlations between *SERPINE1* DNAm and plasma PAI-1 levels in boys and adiposity in girls, no association was observed with plasma PAI-1 levels in girls. As underlined above, the CpG site associated with adiposity in girls is located close to the BX649164 gene initiation start site, which might be regulated by DNAm at this locus but was not quantified in this study (mRNA or protein). Another potential hypothesis might be linked with the Developmental Origin of Health and Disease (DOHaD) concept. Indeed, epigenetic (DNAm) programming is one mechanism likely central to DOHaD which allows for the regulation of gene expression in response to the environment during critical periods of development without modifying the nucleotide sequence [22]. These DNAm adaptations can persist after birth and promote the risk of chronic diseases later in life, such as obesity [32]. Accordingly, the DNAm variation we have identified could have contributed to regulate PAI-1 expression and fat mass accretion earlier in childhood without any more contribution to circulating PAI-1 levels at 5 years of age. Further studies are thus needed to understand when and in response to which environmental stimuli the epigenetic programming of PAI-1 occurs.

## 4. Conclusions

We have identified one CpG site at the *SERPINE1* gene locus where DNAm levels were associated with adiposity in girls at five years of age. We also identified one CpG site at the *SERPINE1* gene locus where blood cell DNAm levels were associated with plasma PAI-1 levels in boys at five years of age. Overall, our results suggest an association between epigenetically mediated regulation at the *SERPINE1* locus and fat mass accretion in girls. This is not supported by an association with plasma PAI-1 levels in girls, which might be possible if adipose tissue expansion in response to higher PAI-1 levels occurred during earlier developmental periods. Longitudinal approaches would therefore be highly relevant and should allow us to investigate this hypothesis. In addition, studies including fine mapping analyses at the *SERPINE1* gene locus as well as quantification of PAI-1 mRNA levels in adipose tissue may help to better understand the molecular mechanisms involved in the associations observed between DNAm and adiposity at 5 years.

## 5. Materials and Methods

### 5.1. Participant’s Selection 

The participants for this study were selected from Gen3G (Genetics of Glucose regulation in Gestation and Growth), a prospective pregnancy and birth cohort described in a previous publication [33]. Briefly, between January 2010 and July 2013, all pregnant women receiving prenatal care and planning to deliver at the Centre Hospitalier et Universitaire de Sherbrooke (CHUS) were invited to participate in Gen3G. A total of 1024 women were recruited during the first trimester of pregnancy. The exclusion criteria were multiple pregnancies, drug and/or alcohol abuse, age less than 18 years, pregestational diabetes, or diabetes diagnosed in the first trimester. In accordance with the Declaration of Helsinki, every mother provided written informed consent before their inclusion in the study. We conducted follow-up visits during the second trimester of pregnancy, at birth, and with the children and mothers approximately 3 years and 5 years after delivery, during which we collected data on multiple phenotypes and biosamples, including anthropometric measures and venous blood samples. For this study, only children with available BMI and DNAm data or plasma PAI-1 levels measurements at 5 years of age were selected (*n* = 341:157 girls and 184 boys). The ethical review board from the CHUS approved the protocol for this study.

### 5.2. Anthropometric Measurement

BMI was calculated by dividing weight by height squared (kg/m^2^). Height (cm) was measured with a stadiometer (Seca, Hamburg, Germany) to the nearest 0.1 cm and weight (kg) was measured with an electronic scale (Rice Lake Weighing systems, Rice Lake, WI, USA). The children were barefoot and were wearing light clothing when height and weight were measured. BMI z-scores (age- and sex-specific) were calculated with the WHO AnthroPlus Software (Reference 2007 for 5–19 years) [34]. Waist circumference (cm) was measured to the nearest 0.5 cm just above the iliac crest, using a Short flip Tape© measuring tape. Skinfold thicknesses (mm) were measured at the biceps, triceps, and subscapular and suprailiac areas to the nearest 0.5 mm, using a skinfold caliper (AMG Medical, Mont-Royal, QC, Canada). Measurements were taken twice, and a third time if the difference between the first two measures was greater than 10%. Total skinfold thickness (mm) was calculated from the sum of the four-skinfold measurement means. The body composition of the children was measured using dual-energy X-ray absorptiometry (DXA) with a Horizon DXA system (Hologic, Marlborough, MA, USA). A trained research assistant ensured that the limbs were in the correct position on the machine and that no metallic objects were present. In addition, children were instructed to remain still while measurements were taken. Hologic software (version 5.5.3.1) was used to define body regions and calculated body fat percentage. The android fat (%), trunk fat (%), and total fat (%) data were used for statistical analysis.

### 5.3. Plasmatic PAI-1 Levels Measurement

Blood samples were collected from the brachial vein from children aged 5 years in a fasting state. Blood plasma was immediately separated from the children’s blood using a laboratory centrifuge and put into aliquots stored at −80 °C until PAI-1 measurements were taken. Plasma PAI-1 levels were then quantified using a multiplexed particle-based flow cytometric assay (Human Milliplex map kits, EMD Millipore). Intra- and inter-assay coefficients of variation were <10% and <15%, respectively.

### 5.4. Measurements of DNAm

Genomic DNA was purified from whole blood samples collected at five years using the AllPrep DNA/RNA/Protein Mini kit (QIAGEN, Hilden, Germany) and DNA concentration was quantified using the Quant-iT^TM^ PicoGreen^TM^ dsDNA Assay Kit (Qiagen, Germantown, MD, USA). Extracted DNA was then bisulfite converted with an EZ-Methylation kit (Zymo Research, Irvine, CA, USA) and analyzed for DNAm using the Infinium MethylationEPIC BeadChip (Illumina, San Diego, CA, USA). Samples were randomly assigned to plates to minimize potential technical bias. Methylation data were imported into R (version 4.1.1) and the *minfi* package [35] was used for preprocessing steps, which include sample and probe quality controls (QCs) [35]. DNAm was measured for 328 samples but only 323 remained available for analysis after excluding two samples for failed QC, one for sex mismatch, and two for single nucleotide polymorphism (SNP) mismatch. During the probe QC procedures, we excluded probes because of nonsignificant detection for more than 5% of the samples (*p* > 0.05). The data were then processed using functional normalization that utilizes the control probes to adjust for technical variability [36]. We then used regression on correlated probes (RCP), a method that uses genomic proximity to adjust the distribution of type 2 probes to counteract technical biases [37]. Based on the Illumina annotation manifest, we selected 15 CpG sites within the *SERPINE1* gene and extracted their DNAm beta-values (ranging from 0 to 1, as an approximation of the percentage of methylation) for further analyses. We performed a logit transformation of the β-values to obtain their M-values, which normalized their distribution [38]. The M-values were then used in the statistical analyses. 

### 5.5. Statistical Analysis

Histograms of each variable were first visually inspected to verify distribution, and normality was assessed using Shapiro–Wilk testing. Variables that did not follow a normal distribution (i.e., plasma PAI-1 level, waist circumference, skinfold thickness, total % fat, android % fat, and trunk % fat) were log_10_ transformed, which attempted to normalize their distribution. Student’s t-tests were applied to test for significant differences (*p* < 0.05) between girls and boys. 

Partial Pearson’s correlations were measured to assess the relationship between plasma PAI-1 levels and adiposity markers at 5 years in all children together and in a model including age and sex as covariates. We then created multivariable regression models with the same covariates as mentioned before to test for an interaction term between plasma PAI-1 levels and sex, assessed for every adiposity marker. 

We then measured partial Pearson’s correlations to assess associations between blood cell DNAm levels (M-values) at each of the 15 selected CpG sites within the *SERPINE1* gene locus and adiposity markers (i.e., BMI z-score, waist circumference, skinfold thickness, total % fat, android % fat, and trunk % fat) and plasma PAI-1 levels. These analyses were firstly conducted on all children together. Associations with adiposity markers as dependent variables were corrected for age, sex, batch effect, and cell-type heterogeneity. The same covariates, in addition to BMI, were used to assess the association between DNAm and plasma PAI-1 levels. We then created multivariable regression models with the same variables as previously described to test for an interaction term between DNAm and sex, assessed for every adiposity marker. CpG sites that showed significant sex interaction terms (p_inter_ < 0.05) were retested in girls and boys separately, with the same covariates as previously described, excluding sex. Associations between DNAm and plasma PAI-1 levels were considered significant at a *p*-value < 0.050. Associations between DNAm levels and adiposity were considered significant at a *p*-value < 0.017 after Bonferroni adjustment for multiple testing, considering three groups of adiposity markers as independent of each other: BMI z-score in addition to central (waist circumference and trunk and android percent fat) and total fat (skinfold and total percent fat) adiposity markers. 

We also measured partial Pearson’s correlations to test for the co-correlation between each of the 15 CpG sites located within the *SERPINE1* gene. These associations were tested in all children together using age, sex, batch effect, and cell-type heterogeneity as covariates and in girls and boys separately with the same covariates as previously described, excluding sex. Statistical analyses were performed with the SPSS software (version 28, IBM SPSS Statistics, Endicott, NY, USA).

## Figures and Tables

**Figure 1 ijms-23-11833-f001:**
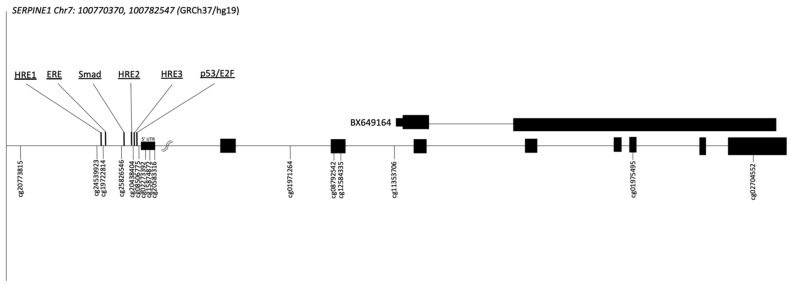
Human *SERPINE1* gene locus and the 15 CpG sites analyzed.

**Figure 2 ijms-23-11833-f002:**
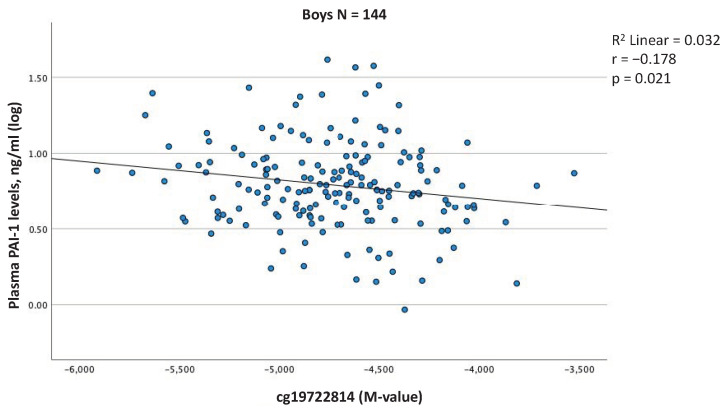
A significant association between blood cell DNAm levels at the *SERPINE1* gene locus (cg19722814) and plasma PAI-1 levels at 5 years of age, in boys. Partial Pearson’s correlation, adjusted for age, batch effect, cell-type heterogeneity, and BMI, was considered statistically significant at *p* < 0.05.

**Figure 3 ijms-23-11833-f003:**
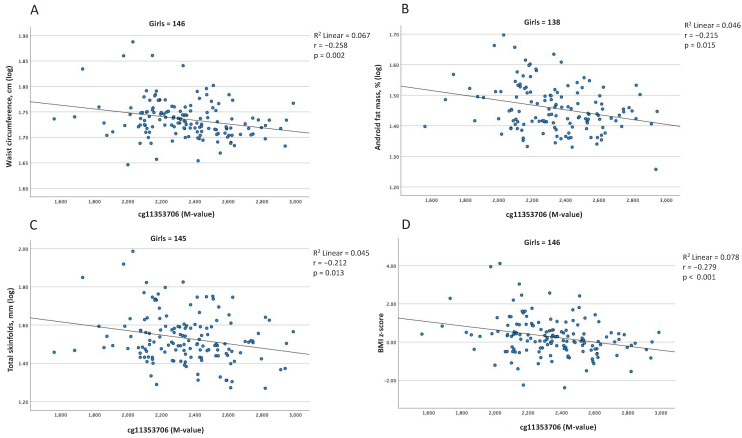
Significant associations between blood cell DNAm levels at the *SERPINE1* gene locus and adiposity levels at 5 years of age, in girls. Associations between DNAm levels at cg11353706 and (**A**) waist circumference, (**B**) android percentage of fat mass, (**C**) total skinfold thickness, and (**D**) BMI z-score. Partial Pearson’s correlations, adjusted for age, batch effect, and cell-type heterogeneity, were considered statistically significant at *p* < 0.017 after Bonferroni correction for multiple testing (i.e., three independent groups of adiposity markers). Abbreviations: BMI, body mass index.

**Table 1 ijms-23-11833-t001:** Characteristics of Gen3G children included in this analysis, stratified by sex.

Characteristics	Girls			Boys			*p* *
	Nb	Mean ± SD	Range	Nb	Mean ± SD	Range	
Age, months	157	64.18 ± 3.83	58.07–79.47	184	64.17 ± 4.01	57.73–86.43	0.982
PAI-1, pg/mL	157	9.99 ± 12.29	1.32–111.87	184	7.84 ± 6.39	0.93–41.33	**0.049**
Weight, kg	157	19.22 ± 3.40	13.10–36.50	184	19.64 ± 2.79	13.00–32.80	0.212
Height, cm	157	109.9 ± 5.2	98.05–129.55	184	112.0 ± 4.9	101.0–126.1	**<0.001**
BMI, kg/m^2^	157	15.82 ± 1.77	12.36–24.15	184	15.59 ± 1.29	12.50–20.64	0.142
BMI z-score	157	0.279 ± 0.971	−2.38–4.120	184	0.188 ± 0.932	−2.57–3.22	0.382
Total skinfold thickness ^a^, mm	155	35.47 ± 11.99	18.63–96.83	184	28.01 ± 7.77	15.67–76.42	**<0.001**
Waist circumference, cm	157	54.61 ± 4.79	44.30–77.20	184	53.65 ± 3.44	45.85–72.50	**0.038**
DXA trunk fat, %	144	28.81 ± 4.60	18.90–45.80	164	23.89 ± 3.03	18.40–33.30	**<0.001**
DXA android fat, %	144	29.03 ± 5.36	18.10–49.80	164	24.11 ± 3.26	17.90–34.80	**<0.001**
DXA total fat, %	144	33.06 ± 4.33	21.90–47.00	164	28.46 ± 3.32	22.30–39.00	**<0.001**

Note: * Student’s t-test. Significant results with *p*-value < 0.05 are in bold. Abbreviations: BMI, body mass index; DXA, dual-energy X-ray absorptiometry; PAI-1, plasminogen activator inhibitor. ^a^ Total skinfold thickness is the sum of biceps, triceps, subscapular, and suprailiac skinfolds.

## Data Availability

The datasets used and/or analyzed during the current study are available from the corresponding author upon reasonable request.

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
