# Peer review of "SERPINE1 DNA Methylation Levels Quantified in Blood Cells at Five Years of Age Are Associated with Adiposity and Plasma PAI-1 Levels at Five Years of Age"

_ijms, 2022, doi:10.3390/ijms231911833_

Round 1

Reviewer 1 Report

Review for International Journal of Molecular Science:

SERPINE1 DNA methylation quantified in blood cell at five years of age are associated with adiposity and PAI-1 plasma levels at five years of age”

Reviewer summary:

In this study, DNA methylation in blood of the SERPINE1 locus was analyzed in children appr. 5 years of age. In parallel, plasma level of PAI-1, which is encoded by SERPINE1, was measured and used for association analyses with anthropometric data as well as the DNA methylation status of SERPINE1. Overall, the study is of interest to the field, especially because PAI-1 level were already associated with obesity development in the literature. However, the study would benefit from additional analyses. Please find comments below.

Major comments:

A figure of the gene structure of SERPINE1 would help the reader to understand what else might be considered in order to explain how DNA methylation at this locus might affect obesity parameters without affecting PAI-1 level directly. The authors mentioned, that BX649164 is located within the SERPINE1 locus and the observed methylation changes might affect this locus as well. If there are data available on gene expression or protein level of this gene, the authors should include association analyses specifically for this region. In general, including measurements of mRNA expression would be beneficial. Instead, functional analysis such as in vitro methylation of SERPINE1 or enzymatic demethylation in cell culture could proof a direct link of this epigenetic mark on the regulation of SERPINE1 expression and PAI-1 protein level. In addition, a hypermethylation of the ERE element was discussed to be of relevance for SERPINE1 level. The authors could use in vitro luciferase assays to eliminate the ERE region and proof functional consequences. In addition, it might be possible to hypermethylate a luciferase construct containing the ERE region and look for consequences on expression level.

In the introduction and discussion, the authors comment on the genomic positions of the analysed CpG sites (promoter, intronic, shelf, shore etc.). The manuscript would benefit from a figure showing the positions of the CpG sites within the genomic context.

As EPIC Arrays were used, I am wondering why only SERPINE1 was the focus of this manuscript. A general genomewide analyses of alterations of DNA methylation in the studied cohort would be possible and of high interest. The same is true for gene expression analyses.

Minor comments:

Including a short notice of the definition of obesity in children would be helpful to readers not being experts in childhood obesity.

In Table S2, please include the reason of multiple testing as cutoff for choosing a p-value < 0.015 for significance.

There are some spelling and grammar errors throughout the manuscript, please check carefully. Here are some recommendations for improvement:

Introduction, line 49: … over the past few decades… …children aged 5 years and below….

Introduction, line 62: please add full version of IRS. …Also, due to its inhibitory effect…..

Introduction, line 64: …the inability of releasing fatty acids…

Introduction, Line 66: please mention, that PAI-039 is an inhibitor of PAI-1.

Introduction, Line 68: …in the development of obesity.

Introduction, Line 73: …not associated with elevated transcription….

Discussion Line 166: …but the impact in gene body…. (please note: although DNA methylation in the promoter region is mostly correlated with mRNA repression, it can also be different. For instance, high methylation leads to high gene expression.)

Author Response

                                                                                                                     September 2022

Dear Ms. Hazel Zhang

We would like to thank you for your interest in our work and the reviewers for their careful revision of our manuscript. Please find enclosed the revised version of our manuscript entitled: “SERPINE1 DNA methylation quantified in blood cells at five years of age is associated with adiposity and plasma PAI-1 levels at five years of age”: changes in the text were highlighted in yellow. You will also find below the responses to the comments raised by the reviewers.

We thank you again and hope this revised version of our manuscript will be considered improved enough for publication in International Journal of Molecular Sciences, special issue Epigenetic Mechanisms and Inflammatory Response in Endocrine Disorders

Best regards,

_______________________

Luigi Bouchard, R.T. Ph.D. MBA
Full professor
Department of Biochemistry and Functional Genomics Université de Sherbrooke
305 St-Vallier Street, Chicoutimi (Québec)
G7H 5H6 Canada

Response to Reviewer 1 Comments

Major comments:

  1. A figure of the gene structure ofSERPINE1 would help the reader to understand what else might be considered in order to explain how DNA methylation at this locus might affect obesity parameters without affecting PAI-1 level directly. The authors mentioned, that BX649164 is located within the SERPINE1 locus and the observed methylation changes might affect this locus as well. If there are data available on gene expression or protein level of this gene, the authors should include association analyses specifically for this region. In general, including measurements of mRNA expression would be beneficial. Instead, functional analysis such as in vitro methylation of SERPINE1 or enzymatic demethylation in cell culture could proof a direct link of this epigenetic mark on the regulation of SERPINE1 expression and PAI-1 protein level. In addition, a hypermethylation of the ERE element was discussed to be of relevance for SERPINE1 level. The authors could use in vitro luciferase assays to eliminate the ERE region and proof functional consequences. In addition, it might be possible to hypermethylate a luciferase construct containing the ERE region and look for consequences on expression level.

Answer #1:  We thank this reviewer for his/her very detailed comment. Although we agree that PAI-1 mRNA levels assessment would improve our manuscript, we believe it would have been unethical to perform an invasive biopsy procedure required for adipose tissue (where PAI-1 is expressed) sampling on young children aged only 5 years old. However, we assessed PAI-1 protein levels (a proxy for mRNA levels) and its association with DNA methylation (DNAm) levels at the SERPINE1 locus. Our results suggest that DNAm might well regulate plasma PAI-1 levels.

The proposed functional analyses are very interesting (do not need adipose tissue biopsies) and highly relevant. However, and before undertaking these complex analyses, we decided first to fine map the DNAm signals we have identified as only tag-CpG sites have been epigenotyped on the EPIC Array. Our approach based on fine mapping will allow to refine the association signals and thus improve the selection of the CpG sites more likely to regulate SERPINE1 expression and obesity development. Depending on the genomic context of these CpG sites, this will also improve the selection of the best model to test these functional hypotheses. 

  1. In the introduction and discussion, the authors comment on the genomic positions of the analysed CpG sites (promoter, intronic, shelf, shore etc.). The manuscript would benefit from a figure showing the positions of the CpG sites within the genomic context.

Answer #2: As suggested, we have included a figure showing the SERPINE1 gene structure (page 4, line 110). We hope that this will help to better understand the results presented in our study.

Figure 1. Human SERPINE1 gene locus and the 15 CpG sites analyzed.

The figure shows the SERPINE1 gene locus, its transcript and the alternative BX649164 transcript (chr7: 100,774,814 [24]). The 15 CpG sites analyzed, and their “cg” identification number are also represented. Exons are shown in black boxes.  Transcription factor binding sites are also shown: three hypoxia-responsive elements (HRE; -452, -195, and -161 from the TSS), an estrogen response element (ERE at -427 to -407), one SMAD protein-binding site mediating TGF-β responsiveness (Smad at -280) and a p53 responsiveness element (P53 at -159 to -134) in the same region of a binding site for PAI-1 negative regulator (E2F) [25].

  1. As EPIC Arrays were used, I am wondering why onlySERPINE1 was the focus of this manuscript. A general genomewide analyses of alterations of DNA methylation in the studied cohort would be possible and of high interest. The same is true for gene expression analyses. 

Answer #3: Although applying a pan-epigenomic approach would be interesting to identify PAI-1 level-associated DNA methylation marks, we have decided to focus on the SERPINE1 gene locus as this region is more likely to harbor epivariations regulating PAI-1 expression. We nevertheless agree that applying an epigenome-wide approach is of interest to identify other (trans) regions potentially regulating PAI-1 expression and we keep this suggestion in mind for our future analyses.

Minor comments:

  1. Including a short notice of the definition of obesity in children would be helpful to readers not being experts in childhood obesity.

Answer #1: This definition has been added in Page 2, lines 51 to 55.

“To classify adiposity levels, the World Health Organization (WHO) recommends the use of the body mass index (BMI), which consists of the measurement of weight in kilograms (kg) divided by the square of the height in meters (m). In children aged 5 to 19 years, WHO recommends the use of BMI z-score which is BMI measure adjusted for age and sex [1]. The threshold for childhood obesity consists of the 97th percentile BMI by age group and sex, which is equivalent to a BMI z-score of 2 [2].”

  1. In Table S2, please include the reason of multiple testing as cutoff for choosing a p-value < 0.015 for significance.

Answer #2: Standard Bonferroni correction for multiple statistical testing is based on the assumption that the tests are independent from each other. Indeed, all the adiposity markers we have analyzed are correlated with each other and thus not independent. To decrease the impact (over correction for multiple testing) of co-correlation between our adiposity markers, we have considered BMI in addition to central (waist circumference and trunk and android percent fat) and total fat (skinfold and total percent fat) adiposity markers. We believe this approach provides an acceptable protection against false positive findings. We added the reason of multiple testing in Table S6 (the table numbers have been modified as a result of changes made in the manuscript) and also corrected the threshold mentioned (p <0.017 (0.05/3) and not p <0.015 as written). We are sorry about this “typo”.

  1. There are some spelling and grammar errors throughout the manuscript, please check carefully. Here are some recommendations for improvement.

Answer #3: We thank you this reviewer for paying attention to the spelling errors. The manuscript has been carefully reviewed, including for all specific recommendations.

  1. Discussion Line 166: …but the impact in gene body…. (please note: although DNA methylation in the promoter region is mostly correlated with mRNA repression, it can also be different. For instance, high methylation leads to high gene expression.)

Answer #4: Thank you for underlining this clarification. We have reviewed the manuscript accordingly.

Reviewer 2 Report

This manuscript by Taschereau et al. investigates the cross-sectional relationship of DNA methylation of the SERPINE1 in blood, PAH-1 levels, and adiposity measures in children at 5 years of age. At a small number of CpG sites, they identified sex-specific associations between DNA methylation and PAH-1 levels or adiposity measures. Sex differences in the development of obesity risk is an important topic, but this manuscript lacks sufficient clarity throughout to allow the reader to assess the importance of these findings. The cross-sectional nature of this study also means it is unable to address or contribute to the question of causality that is raised throughout the introduction and discussion sections.

Comments:

1. The authors have used an inconsistent approach to using p-value significance thresholds adjusted for multiple tests. An adjusted threshold has been used for the 3 'categories' of adiposity measures, but there's no adjusted threshold for the number of CpG sites tested. As the authors demonstrate the DNA methylation level each of the SERPINE1 CpGs is only weakly correlated, having an association of p<0.05 for 1 of the CpG sites due to chance alone is likely, as is the case for the reported sex-interaction effect between DNA methylation and PAH-1 levels. The authors need to be considerably more cautious in there interpretation of this 'significant' association, and include this in the discussion.

2. The authors' hypothesis of DNA methylation influencing PAH-1 levels which in turn may contribute to the development of obesity does not appear to be supported by the findings from this study. As the abstract and introduction both detail this hypothesis as the rationale for the study, the lack of association between PAH-1 levels and adiposity measures seems like a critical detail to include in the abstract and the conclusion of the manuscript, but it is currently absent from both. 

3. It would be very helpful for the authors to include a supplementary figure showing the genomic position of each of the SERPINE1 CpG sites investigated in this study relative to the promotor region and gene bodies early in the results section. This would provide more context for the CpG sites as they are mentioned in the results.

4. Relatedly, it is unclear why the correlation of DNA methylation levels between each CpG site is presented as the final section of the results. This would be more helpful to present early in the results to give more context for similarities or differences in findings across CpG sites. 

5. This may be just be the case in the file provided to the reviewers, but Figures 1 and 2 both appear to be of low graphical quality, making it hard to read the text and distinguish the details.

Author Response

                                                                                                                     September 2022

Dear Ms. Hazel Zhang

We would like to thank you for your interest in our work and the reviewers for their careful revision of our manuscript. Please find enclosed the revised version of our manuscript entitled: “SERPINE1 DNA methylation quantified in blood cells at five years of age is associated with adiposity and plasma PAI-1 levels at five years of age”: changes in the text were highlighted in yellow. You will also find below the responses to the comments raised by the reviewers.

We thank you again and hope this revised version of our manuscript will be considered improved enough for publication in International Journal of Molecular Sciences, special issue Epigenetic Mechanisms and Inflammatory Response in Endocrine Disorders

Best regards,

_______________________

Luigi Bouchard, R.T. Ph.D. MBA
Full professor
Department of Biochemistry and Functional Genomics Université de Sherbrooke
305 St-Vallier Street, Chicoutimi (Québec)
G7H 5H6 Canada

Response to Reviewer 2 Comments

Comments:

  1. The authors have used an inconsistent approach to using p-value significance thresholds adjusted for multiple tests. An adjusted threshold has been used for the 3 'categories' of adiposity measures, but there's no adjusted threshold for the number of CpG sites tested. As the authors demonstrate the DNA methylation level each of the SERPINE1 CpGs is only weakly correlated, having an association of p<0.05 for 1 of the CpG sites due to chance alone is likely, as is the case for the reported sex-interaction effect between DNA methylation and PAH-1 levels. The authors need to be considerably more cautious in there interpretation of this 'significant' association, and include this in the discussion.

Answer #1: Standard Bonferroni correction for multiple statistical testing is based on the assumption that the tests are independent from each other. Indeed, all the adiposity markers we have analyzed are correlated with each other and thus not independent. This is also true for the epivariants (CpG sites) we have tested. For this study, we have decided to focus on adiposity markers to adjust for multiple testing. Therefore, we have adjusted our results for 3 “groups” of adiposity markers: BMI in addition to central (waist circumference and trunk and android percent fat) and total fat (skinfold and total percent fat) adiposity markers. Focussing on epivariants wouldn’t have change significantly our results as our 2 best association signals would have remained significant after the more stringent correction needed (0.05/15 epivariants = <0.003). We thus believe our approach provides an acceptable protection against false positive findings. We nevertheless agree that this is very important and revised the manuscript accordingly.

  1. The authors' hypothesis of DNA methylation influencing PAH-1 levels which in turn may contribute to the development of obesity does not appear to be supported by the findings from this study. As the abstract and introduction both detail this hypothesis as the rationale for the study, the lack of association between PAH-1 levels and adiposity measures seems like a critical detail to include in the abstract and the conclusion of the manuscript, but it is currently absent from both. 

Answer #2: This reviewer is right, the lack of association between PAI-1 levels and adiposity does not support our hypothesis. However, PAI-1 may have act locally (e.g. adipose tissue) and/or at specific (earlier) developmental periods, which couldn’t be detected in our study. We nevertheless believe the associations between DNAm marks at the SERPINE1 gene locus and adiposity support indirectly this hypothesis. The abstract and Discussion were revised accordingly.

Abstract (Page 1, lines 43 to 45): “Our study suggests that DNAm levels at the SERPINE1 gene locus are negatively correlated with adiposity in girls only, but not with plasma PAI-1 levels, in young girls.”  

Conclusion (Page 9, lines 217 to 220): “This is not supported by an association with plasma PAI-1 levels in girls, which might be possible if adipose tissue expansion in response to higher PAI-1 levels occurred during earlier developmental periods. Longitudinal approaches would therefore be highly relevant and should allow to investigate this hypothesis.”

  1. It would be very helpful for the authors to include a supplementary figure showing the genomic position of each of the SERPINE1 CpG sites investigated in this study relative to the promotor region and gene bodies early in the results section. This would provide more context for the CpG sites as they are mentioned in the results.

Answer #3: As suggested, we have included a figure showing the SERPINE1 gene locus (page 4, line 110). We hope that this will help to better understand the results presented in our study.

Figure 1. Human SERPINE1 gene locus and the 15 CpG sites analyzed.

The figure shows the SERPINE1 gene locus, its transcript and the alternative BX649164 transcript (chr7: 100,774,814 [24]). The 15 CpG sites analyzed, and their “cg” identification number are also represented. Exons are shown in black boxes.  Transcription factor binding sites are also shown: three hypoxia-responsive elements (HRE; -452, -195, and -161 from the TSS), an estrogen response element (ERE at -427 to -407), one SMAD protein-binding site mediating TGF-β responsiveness (Smad at -280) and a p53 responsiveness element (P53 at -159 to -134) in the same region of a binding site for PAI-1 negative regulator (E2F) [25].

  1. Relatedly, it is unclear why the correlation of DNA methylation levels between each CpG site is presented as the final section of the results. This would be more helpful to present early in the results to give more context for similarities or differences in findings across CpG sites. 

Answer #4: We agreed with this reviewer and as suggested, we added a section at the beginning of the results presenting the SERPINE1 gene, the genomic context of its 15 annotated CpG sites as well as the correlation of their DNAm levels (page 3, lines 105 to 109).

The DNAm correlation between adiposity associated CpG sites were more detailly presented in the section reporting these associations (page 5, lines 156 to 161).

  1. This may be just be the case in the file provided to the reviewers, but Figures 1 and 2 both appear to be of low graphical quality, making it hard to read the text and distinguish the details.

Answer #5: The graphical quality of Figures 2 and 3 (note that due to changes in the manuscript, the figure numbers have been changed) has been improved as suggested.

Round 2

Reviewer 2 Report

The authors have clearly addressed my previous comments. I am satisfied that the conclusions of the study are more clearly articulated. The revisions to the results section have made the findings easier to interpret.